

# Generative artificial intelligence in topic-sentiment classification for Arabic text: a comparative study with possible future directions

Fatima Alderazi[1], Abdulelah Algosaibi[1], Mohammed Alabdullatif[1], Hafiz Farooq Ahmad[1], Ali Mustafa Qamar[2] and Abdulaziz Albarrak[3]

[1] Computer Science Department, College of Computer Sciences and Information Technology (CCSIT), King Faisal University, Al-Ahsa, Saudi Arabia
[2] Department of Computer Science, College of Computer, Qassim University, Buraydah, Saudi Arabia
[3] Department of Information Systems, College of Computer Sciences and Information Technology (CCSIT), King Faisal University, Al-Ahsa, Saudi Arabia

Corresponding author
Fatima Alderazi,
220001672@student.kfu.edu.sa

## ABSTRACT

Social media platforms have become essential for disseminating news and expressing individual sentiments on various life topics. Arabic, widely used in the Middle East, presents unique challenges for sentiment analysis due to its complexity and multiple dialects. Motivated by the need to address these challenges, this article develops methods to overcome the lack of topic-based labeling techniques, compares different approaches for preparing extensive, annotated datasets, and analyzes the efficacy of machine learning (ML), deep learning (DL), and large language models (LLMs) in classifying Arabic textual data. Our research utilizes the topic-modeling technique to build a topic-based sentiment dataset of Arabic texts aimed at enhancing our understanding and processing capabilities. We present a comprehensive evaluation of dataset balancing techniques, including under-sampling, over-sampling, and using imbalanced datasets, providing insights into how these approaches impact classification outcomes. Additionally, we explore the influence of dataset sizes on the performance of various ML models, highlighting the importance of dataset scale in developing effective Arabic NLP applications. A further focus of our study is the comparative analysis of generative artificial intelligence (AI) models, including the emerging ChatGPT LLM, assessing their effectiveness in managing the complexities of Arabic language classification tasks. Our results show that support vector machines (SVM) achieved the highest performance, with F1-scores of 0.97 and 0.96 in classifying sentiment and topic, respectively, in Arabic tweets. This research not only benchmarks existing methodologies but also paves the way for more nuanced and robust models in the future, enhancing the application of generative AI in Arabic topic-based sentiment analysis.

# INTRODUCTION

Since the World Health Organization (WHO) announced COVID-19 as a pandemic at the beginning of 2020, social media users have increased, reaching 4.48 billion people worldwide (*Backlinko, 2021*). Under the influence of various life events, people utilize social media platforms to express and share their thoughts and opinions. Twitter, a widely used social media platform, had 396.5 million active users in the fourth quarter of 2021.

Due to the pandemic, governments and organizations started to consider using social media to measure and monitor public opinion related to pandemic events and to put new regulations in place to defend against the impacts of the virus. Several studies have examined the effects of social media on people during COVID-19 (*Alhajji et al., 2020*; *Alnasser et al., 2020*).

Arabic is the world's fifth most spoken language, consisting of 12.3 million words. It is the official language of 22 countries and is spoken by more than 400 million people. However, there are several challenges associated with analyzing the Arabic language. For example, people mostly use Modern Standard Arabic (MSA) for communication on social media, and the dialects used in social media posts are mutually incomprehensible. Another factor that makes it more challenging to classify Arabic tweets is that they are typically short and include platform-specific elements like hashtags and user mentions.

Social media posts are usually written in an unstructured way in natural languages. Natural language processing (NLP) is utilized to analyze the enormous amount of published text by applying machine learning (ML) and Artificial Intelligence (AI) techniques that can analyze data patterns to deduce valuable meanings (*Arora, Banerjee & Narasu, 2020*). Sentiment Analysis (SA) is one of the NLP techniques that examines and explores the subjectivity of data from unstructured content by implementing several well-known machine learning algorithms (*Mohri, Rostamizadeh & Talwalkar, 2018*). Another technique in NLP is topic classification, which examines the co-occurrences of terms in a text and assigns the text to one of several pre-determined topics. Using machine learning algorithms, automated topic classification can classify text into specified categories. Topic classification determines the general topic, whereas SA merely determines the subjective sentiment of a text, but having a topic-based SA approach helps in getting more profound insights into the text by analyzing a text's sentiment concerning a broad topic, such that a text has a positive, negative, or neutral feeling regarding politics, business, or health topics. Topic-based SA combines two NLP fields, and this combination helps get valuable information regarding a particular event. A more human-centered approach may be applied in several industries by using topic-based sentiment classification to public posts, including brand management, employee engagement, and healthcare. Analyzing the sentiment and determining the text's topic could help learn much about individuals' thoughts. For example, when users comment on an event, they usually do so on an event's topic. This comment does not imply that their overall opinion of the event is good or bad; users typically leave feedback regarding different event aspects in both positive and negative sentiments.

When working with a tweet-based dataset, an imbalanced dataset is a prevalent problem that needs to be addressed. An imbalanced dataset is one in which the majority class is substantially more significant than the minority class (*Baron, 2020*). Many techniques for balancing the data distribution could be employed to reduce the imbalanced dataset's impact on the model's learning process. One is oversampling, which replicates minority class examples randomly to produce an equal distribution of classes. The Synthetic Minority Oversampling Technique (SMOTE) is one of the most widely utilized oversampling approaches for resolving the problem of an imbalanced dataset (*Chawla et al., 2002*). Another balancing technique is based on the under-sampling of the majority class.

This research exclusively focuses on enhancing Arabic NLP by developing a novel approach for topic-based sentiment analysis. Specifically, the study investigates the utility of unsupervised topic modeling techniques, primarily the Latent Dirichlet Allocation (LDA), for automatic labeling in creating a robust, multi-labeled dataset suitable for Arabic texts. This work substantially extends our previously published conference papers; in the first one, we proposed building a framework that integrates the two fields: Sentiment Analysis and Topic Classification (*Alderazi, Algosaibi & Alabdullatif, 2022*), the second paper provided an up-to-date literature review, presented the LDA results and used it for topic annotation and showed different models' experimental results on the dataset (*Alderazi, Algosaibi & Alabdullatif, 2021*).

The primary objective of this study is to explore the efficacy of unsupervised topic modeling, particularly through LDA, as a tool for generating labeled datasets for Arabic text. Secondary objective include the application of this labeled dataset to evaluate and enhance the performance of various sentiment analysis and topic classification models tailored for Arabic texts. The main research question guiding this study is: 'How can unsupervised topic modeling technique, be effectively utilized as a labeling tool to generate a reliable multi-labeled dataset for Arabic NLP applications?' This question addresses the core focus of the research, aiming to bridge the gap between topic classification and sentiment analysis within the context of Arabic language content.

In this article, our major contributions to Arabic NLP and topic-based sentiment analysis are distinctly framed around comparative analyses of various approaches. Specifically, we present:

1. A comprehensive evaluation of dataset balancing techniques: We systematically analyze the impact of different dataset balancing techniques (under-sampling and over-sampling) and using imbalanced datasets on the performance of each model. This analysis provides insights into how balancing techniques affect model outcomes on the developed multi-label dataset.

2. In-depth analysis of model performance across dataset sizes: We examine how different sizes of datasets influence the performance of both ML and long short term memory (LSTM) models. This contribution is significant as it benchmarks the dataset size's impact on model accuracy, which is critical for developing efficient Arabic NLP applications.

3. Comparative analysis of generative AI in Arabic topic-based classification: To evaluate the effectiveness and applicability of generative models in handling the intricacies of Arabic language classification tasks.

The article is organized in the following manner: "Literature Review" provides a brief literature review, and "Dataset" presents the dataset in detail. Later, "Methodology" shows the methodology; "Machine Learning (ML)" presents the experiments performed with ML algorithms. Similarly, "Deep Learning (DL)" presents the experiments performed with DL. In contrast, "Generative AI" presents the results of applying LLMs, "Results Discussion" discusses and analyzes the obtained outcomes from ML, DL, and LLMs, then compares them to similar works, "Future Direction" provides some future directions of our research, and the last section concludes the paper.

## LITERATURE REVIEW

Arabic is one of the official United Nations languages (*United Nations, 2022*), and several datasets exist for Arabic NLP work. One such work has been done for sentiment analysis by *Mourad & Darwish (2013)*, where they extracted the sentiment present in a dataset of 2,300 tweets using the Naïve Bayes (NB) classifier. Their experimental setup used various preprocessing steps, including bi-grams, part-of-speech tagging, and stemming. It achieved 76.6% accuracy for determining a tweet's subjectivity and reached 80.5% accuracy for polarity detection. *Liu, Zhong & Guo (2013)* employed multiple classifiers, specifically the SVM and NB classifiers, for sentiment analysis. A total of 500 Arabic-language film reviews have been collected to build the dataset. The SVM classifier's accuracy was 90%, whereas NB's accuracy was 84%.

*Chakraborty et al. (2020)* utilized a dataset with a thousand tweets divided into 500 positives and 500 negatives. They concentrated on sentiment analysis at the sentence level, and the SVM and NB were employed. To examine the effects of dialectical Arabic on performance, the authors added a few Egyptian terms to Modern Standard Arabic. With an accuracy of 72.6%, the data demonstrate that SVM outperformed the NB classifier in sentiment analysis. *Al-Rubaiee et al. (2016)* conducted research to classify Arabic tweets based on their sentiment as positive, negative, or neutral regarding e-learning. They began by collecting and preprocessing tweets, which resulted in 1,121 tweets. The tweets were then carefully labeled with their sentiment. They used SVM and NB as classification methods, TF-IDF weighting methods, and N-grams for feature extraction. After that, they conducted two experiments with two separate classes. Positive and negative categories were used in the first experiment, whereas positive, negative, and neutral were used in the second. SVM got the best results and achieved an accuracy of 84.84% in the first experiment and 73.15% in the second.

*Al-Horaibi & Khan (2016)* used ML techniques (SVM and NB) to investigate Arabic Sentiment Analysis (ASA) in e-learning by analyzing tweets related to King Abdulaziz University. Their dataset contains 2,000 tweets, and the accuracy of deep learning classifiers reached 81%. In addition, the authors propose using a fuzzy rule framework based on a Gaussian membership function to identify the tweets' sentiments correctly. This

approach achieved 79% accuracy. *Alruily & Shahin (2020)* proposed a sentiment analysis system based on two models for analyzing tweets about Saudi universities. The first model is SVM, while the second is an ensemble of multiple classifiers. They concluded that SVM outperformed the ensemble model, achieving 93.52% accuracy. The authors in *Al-Laith et al. (2021)* introduced AraSenCorpus, a semi-supervised self-learning methodology with 4.5 million tweets, including MSA and several Arabic dialects. *Alahmary, Al-Dossari & Emam (2019)* used two deep learning algorithms to analyze the sentiment of a dataset of 32,063 tweets: LSTM and bidirectional LSTM (Bi-LSTM). *Mohammed & Kora (2019)* presented deep learning (DL) models for Arabic SA on 40K Arabic tweets, specifically CNN, Region-based Convolutional Neural Network (RCNN), and LSTM models. The data underwent preprocessing before being used to classify texts with DL models. According to the experimental findings, LSTM outperforms CNN and RCNN with an average accuracy of 81.31%.

Additionally, adding data to the *corpus* improves LSTM accuracy by 8.3%. As part of their investigation into the viability of using one-way analysis of variance (ANOVA) as a feature selection method to significantly reduce the number of features for Arabic tweets opinion classification, the authors in *Alassaf & Qamar (2022)* built a dataset of approximately 8,144 tweets related to Qassim University in Saudi Arabia. Different classifiers have been used in the experiment. With an F1 score of 69%, SVM and NB are considered the best-performing classifiers.

A study was done by *Alhajji et al. (2020)* to measure the impact of seven COVID-19 preventions. To analyze their sentiment, they used the Natural Language Toolkit (NLTK) module in Python by applying the NB model to Arabic tweets containing hashtags related to seven government-imposed public health initiatives. The analysis of 53,127 tweets revealed more positive than negative tweets. *Mubarak & Hassan (2021)* built an Arabic dataset of 8,000 COVID-19-related tweets manually annotated that classified the tweets into 13 classes. The dataset has been evaluated using various ML classifiers. According to the findings, SVM showed the best performance, achieving 85.4% for binary classification and 62.8% for fine-grained classification. *Alsudias & Rayson (2020)* clustered a collection of one million Arabic tweets on COVID-19 published between December 2019 and April 2020 into five categories: statistics, prayers, disease locations, advice, and advertising. This research applies a rumor detection approach by analyzing 2,000 arbitrary tweets referring to the Saudi Arabian Ministry of Health. *Abdelgwad et al. (2022)* designed and implemented an Arabic tweet sentiment analysis regarding distance learning in Saudi Arabia. They gathered and preprocessed two separate datasets from tweets published from several locations in Saudi Arabia between March 2020 and April 2021. The analysis found that the accuracy rate decreases as the N-gram size increases.

*Abuzayed & Al-Khalifa (2021)* present a BERT-based Arabic SA and sarcasm detection technique. This article worked with seven BERT-based models and supplemented the shared task data collection to classify tweets' sentiment or recognize sarcasm, achieving the best performance of 0.65 F1-score with the MARBERT model. As an alternative NLP technique, some datasets have been created for topic classification. *El-Halees (2011)*, for instance, has gathered 1,143 posts on three different subjects: politics, sports, and

education. There are 8,793 sentences in these posts. The author classifies the documents using three classifiers in sequence to boost accuracy. The total average accuracy was 80%. *Aldayel & Azmi (2016)* proposed a lexical-based classifier data labeling. The generated labeled data are used to train the SVM classifier. The results of the studies demonstrate that this hybrid strategy increased the lexical classifier's F-measure by 5.76%. With a 16.41% accuracy improvement, the overall F-measure and accuracy values were 84% and 84.01%, respectively. *Bekkali & Lachkar (2019)* introduced topic modeling as a method for bringing together terms with similar semantic linkages based on the idea that terms belonging to the same topic share numerous semantic links in the same dataset, and their related concepts will have the same semantic relations in the same dataset. The proposed approach was tested and validated using the large-scale Arabic book reviews dataset. Using the NB and SVM classifiers, the effectiveness of the proposed system has been evaluated in terms of the F1 score. The findings show that NB acquired the greatest F-measure of 81%, while using partial conceptualization with LDA.

In *Abdelrazek et al. (2022)*, they applied unsupervised machine learning to identify latent topics in Saudi newspapers. The authors used the Latent Dirichlet Allocation (LDA) algorithm with seven identified topics. These topics include surveillance, development, sports, health, economics, domestic affairs, and international politics. The seven-topic model achieved a coherence degree of 0.6723, while the qualitative evaluation involved reviewing the coherence of each topic's words.

Article (*Beseiso, 2019*) presents a new approach to extract lexical features based on semantic aspects (topics) using LDA for sentiment analysis of Arabic tweets. The proposed model shows better results than different algorithms, such as SVM and NB, making it a better approach for sentiment analysis for the described data.

Only some works have focused on building multi-labeled data to study Arabic topic-based sentiment analysis. The researchers demonstrated in *Omar et al. (2021)* that by manually annotating a multi-label Arabic dataset, a semi-supervised annotation technique might be applied to multi-label classification, sentiment analysis, and short-text classification. It consists of 11 balanced classes comprising 44,000 posts from Facebook and Twitter. The topic classification, sentiment analysis, and multi-label classification experiments have been done using nine ML classifiers with three feature representations. With parameter tuning and chi-square feature selection, the Linear Support Vector Classification (SVC) classifier with N-gram (*TWI, 2015*; *Google Sheets, 2023*) outperformed other classifiers with an accuracy of 97.92%. Furthermore, the author collected tweets in *Boujou et al. (2021)* to build an open-source multi-label dataset. This dataset contains over 50K tweets in five national dialects: Algerian, Lebanese, Moroccan, Tunisian, and Egyptian. It also includes three sentiment classes—Positive, Negative, and Neutral—and five topic labels—Other, Politics, Health, Social, Sport, and Economics. This dataset has been tested using a variety of classifiers, including the Stochastic Gradient Descent (SGD) Classifier, Logistic Regression, NB, and Linear SVC. SGD performed with the highest sentiment analysis accuracy (77%), whereas Linear SVC achieved the best topic classification accuracy (84%) and dialect allocation accuracy (76%).

In using Generative Pretrained Transformer (GPT) models in sentiment analysis and their departure from current machine learning approaches. *Cui et al. (2021)* demonstrates the superiority of GPT-based strategies in sentiment analysis, achieving a significant increase in F1-score compared to state-of-the-art models.

*Mohamed, Ali & Madbouly (2021)* proposes an aspect-level sentiment analysis model that combines GPT and multi-layer attention, resulting in improved accuracy compared to other models. *Zhou et al. (2021)* explores the effectiveness of using synthetic text datasets generated by GPT-3 for sentiment analysis, showing that deep learning models trained on these datasets achieve higher accuracy. Overall, these papers emphasize the potential and advantages of GPT models in sentiment analysis, showcasing their advanced performance and departure from traditional machine learning methods.

In our work, we study the topic-modeling technique to automate the labeling of an Arabic dataset to fill the gap in the manual annotation process observed in previous work for building multi-label datasets. This process is time and effort-consuming and requires the availability of linguistics with each dialect when planning to generalize the dataset. We later compared the experiment results for several ML modules and ChatGPT to evaluate the dataset performance scientifically and comprehensively analyzed the generative AI capability on topic-based sentiment analysis.

## DATASET

Social media platforms are a targeted resource for analyzing people's opinions on a specific event. Therefore, the enormous amount of data that Twitter makes available is a valuable tool for creating NLP applications. This article utilized our previously generated multi-label dataset for topic-based sentiment classification (*Alderazi, 2021*). The dataset is an updated version of a dataset produced for sentiment analysis applications by the King Abdullah University of Science and Technology (KAUST) (*Alharbi et al., 2020*). This dataset has 95 K tweets with positive, negative, and neutral categories. It is an extensive, high-quality, manually annotated Arabic dataset. The dataset is a general Arabic dataset that does not target a single dialect. It was collected using Twitter's public streaming API between May 2012 and April 2020. It would be an excellent choice for a generalized Arabic NLP application. In addition, in our previous work (*Alderazi, Algosaibi & Alabdullatif, 2022*), we worked on a subset of this dataset to annotate the tweets with their related topics and generate the multi-label version. The tweets have been annotated with topic classes using the Latent Dirichlet allocation (LDA) method. One of the most popular methods for probabilistic text modeling in machine learning is LDA (*Campbell, Hindle & Stroulia, 2015*; *Intisar et al., 2019*; *Yau et al., 2014*). LDA utilization in Arabic has been proven in *Zrigui et al. (2018)*.

LDA is an unsupervised machine learning technique used to recognize the latent topic structure of textual documents. LDA assumes that each dataset is a random mixture of latent topics, and each latent topic is characterized by a word distribution. Figure 1 shows the outcome of defining the five most occurring topics in the dataset, as well as Fig. 2 shows the words related to each topic (*e.g.*, Universities, Corona, COVID, Wuhan City, Alsalam Alikum, cases).

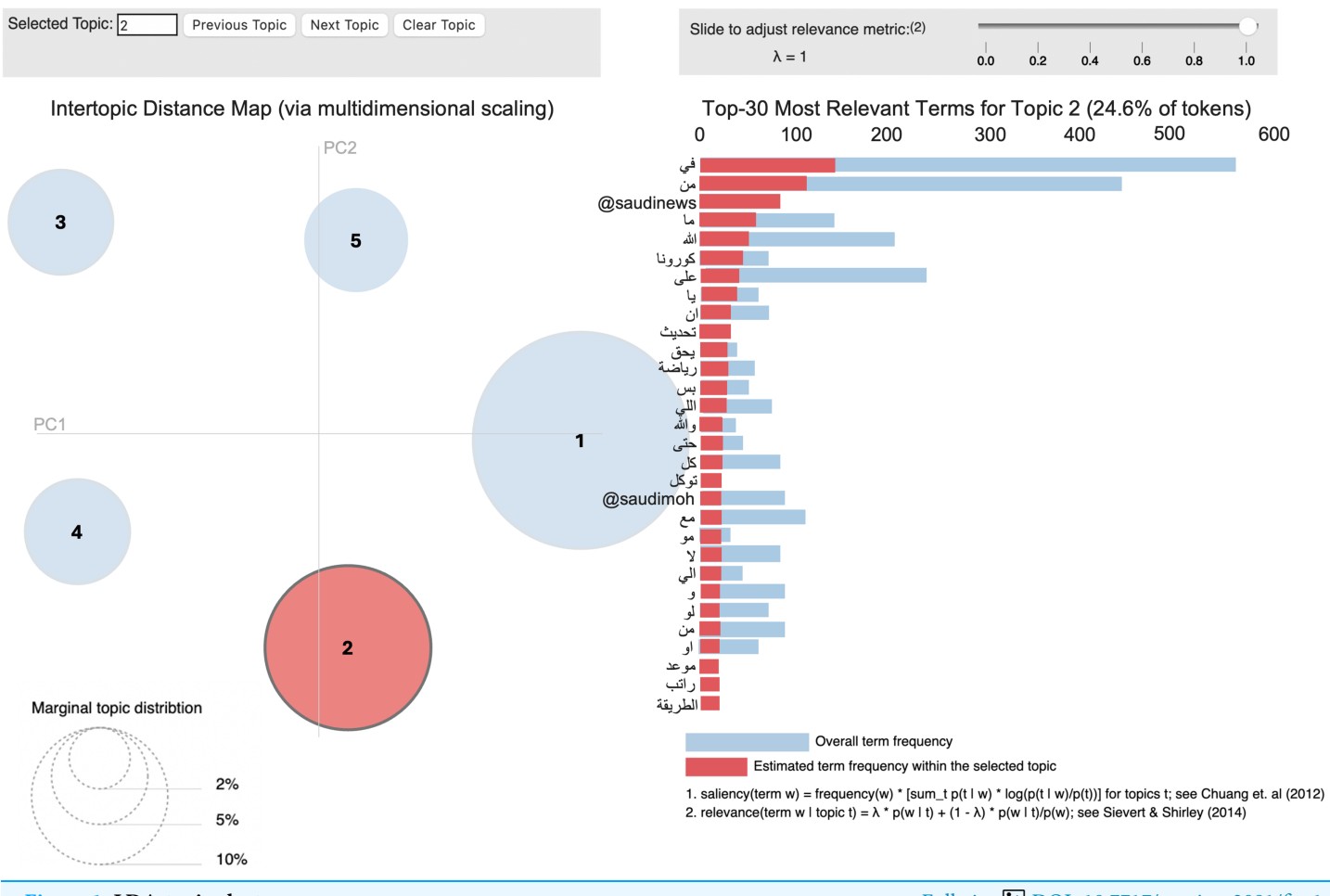

**Figure 1** LDA topic clusters.

In this study, the gensim library was used to create bi-gram representations of the text and to run LDA. Gensim's LDA implementation requires text to be presented as a sparse vector. Conveniently, gensim also provides utilities to convert NumPy dense matrices or scipy sparse ones into the required form.

Moreover, the coherence score achieved by the LDA model was 0.029. Although this might appear low at first glance, it is essential to consider the specific context and nature of the content being analyzed. The dataset consists of MSA Tweets, which are inherently unstructured and diverse in topics. MSA Tweets can include informal language, regional dialects, abbreviations, and non-standard use of language, making it challenging to achieve high coherence in topic modeling. This indicates that, despite the complexity and diversity of the data, the LDA model was able to capture the underlying topic structures within the MSA Tweets.

To further ensure the model's reliability, we have manually verified the produced topic labels. This step involved reviewing the topics and their corresponding word clouds to

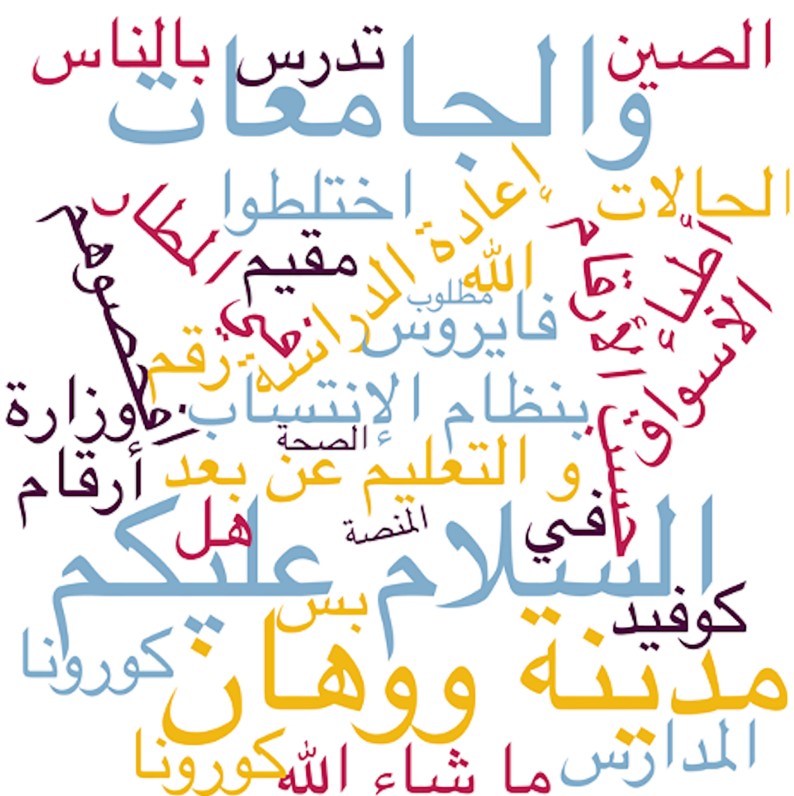

**Figure 2** **Most frequent words cloud.**

confirm that the topics identified by the LDA model are meaningful and relevant (*Alderazi, 2021*).

A multi-labeled dataset allows consumers to understand the data better. We can elicit popular opinions on a given topic. Analyzing two classes together can be used in various situations, such as monitoring social media by examining posts that frequently contain some of the most candid opinions about products, services, and businesses. Moreover, due to the variety of topics and many departments within a firm, it automatically processes customer service/feedback from emails, chats, phone call records, and helpdesk tickets. Furthermore, it facilitates product analysis to determine how the target audience regards a product and which aspects need to be enhanced immediately after launch or after evaluating years of customer input. The classification of three topics—politics, business, and health—has been the primary objective of this research. 32K+ Arabic Tweets with sentiment and topic classes form the updated dataset. Each tweet has been annotated with the most associated topic using its word probability representation.

As shown in Tables 1 and 2, the dataset has an imbalanced distribution, as indicated by the majority of natural sentiment labels in the sentiment class and the Health label in the topic class. This sentiment-neutral majority in the Twitter dataset has been studied by *Chakraborty et al. (2020)*. Data balancing techniques should be applied as part of the preprocessing steps to minimize the effect of the imbalanced dataset. Two data balancing techniques have been applied: oversampling and under-sampling. SMOTE algorithm has

**Table 1 Classes samples distribution.**

| Class | Label | Number of samples |
|---|---|---|
| Sentiment | Positive | 5,034 |
| | Negative | 4,930 |
| | Neutral | 22,800 |
| Topic | Politics | 4,915 |
| | Health | 22,260 |
| | Business | 5,588 |

**Table 2 Sample distribution for topic-based sentiment.**

| Sentiment label | Topic label | Number of samples |
|---|---|---|
| Positive | Politics | 167 |
| | Health | 4,596 |
| | Business | 272 |
| Negative | Politics | 327 |
| | Health | 3,499 |
| | Business | 1,106 |
| Neutral | Politics | 4,423 |
| | Health | 14,167 |
| | Business | 4,212 |
| Total | | 32,769 |

been implemented as an over-sampling technique. It is one of the most commonly used methods to solve the imbalanced problem. The algorithm operates by randomly picking an instance from the minority class and identifying its k closest neighbors. A synthetic instance is subsequently formed by randomly selecting one neighbor (b) from the k closest ones and then establishing a line segment in the feature space by linking instances (a) and (b). The synthetic instances arise through a convex mixture of the two selected instances (a) and (b) (*Al-Horaibi & Khan, 2016*).

## METHODOLOGY

In tweet sentiment analysis, the key task is to classify the tweets as positive, negative, or neutral. Whereas for topic classification, the essential process is based on sorting the content into one of the pre-established categories. The original KAUST dataset has been published in a sentiment analysis competition to test its performance. The best results achieved a 0.849 F1-score by applying a dense layer on top of MARBERT. This study proposes annotating tweets using the LDA algorithm for topic modeling to generate a multi-label dataset for more accurate topic-based sentiment classification of tweets. It also proposed a detailed comparison between several ML and DL models that investigate the benefits of a multi-label dataset for the Arabic topic-based sentiment analysis of tweets.

Sentiment analysis and topic classification problems rely on well-known machine learning techniques on text (*Raschka & Müller, 2021*). A survey conducted by *Alrefai, Faris & Aljarah (2018)* found that the most utilized supervised classification algorithm in Arabic sentiment analysis is the SVM. In this study, in addition to SVM, we have employed three other machine learning classifiers, namely naive Bayes (NB), K-nearest neighbors (KNN), and logistic regression (LR), to provide a comprehensive comparison and to validate the effectiveness of different approaches in sentiment analysis.

Furthermore, modifications were made to the ML parameters to enhance the model's performance and accuracy:

1. The Term Frequency-Inverse Document Frequency (TF-IDF) technique was applied for word vectorization to preserve the order of the labels each time.
2. The training and test texts were tokenized using the same tokenizer (TweetTokenizer), maintaining consistency and reducing discrepancies between the training and testing phases.

Deep neural networks have the capability to model sophisticated abstractions and decrease dimensionality by utilizing multiple processing layers based on complex configurations or non-linear transformations. The architecture of the LSTM is devised in a way that makes it a potent method for addressing the vanishing gradient dilemma found in Recurrent Neural Networks (RNN) (*Zhao et al., 2017*). The LSTM architecture, a type of recurrent neural network, employs LSTM cell blocks instead of conventional neural network layers and is usable for modeling sequential data. The choice of using LSTM as the DL algorithm was influenced by previous studies (*Alahmary, Al-Dossari & Emam, 2019*; *Mohammed & Kora, 2019*), which utilized DL algorithms for NLP tasks using both CNN and LSTM algorithms. Nevertheless, LSTMs demonstrated superior performance in the majority of the examined instances.

The classification of MSA tweets by topic and sentiment is one of the main objectives of this article. We deployed an unsupervised topic modeling technique to build a multi-labeled dataset using the sentiment class and label the dataset with the topic class.

The method that has been followed to build the topic-based sentiment classification approach is shown in Fig. 3. The system has six main phases:

1. Data collection: Arabic tweets are fetched through the Twitter API.

2. Apply preprocessing steps to the dataset: Text normalization, data cleaning, tokenizing, and balancing of class distribution.

3. Topic generation: Applying the LDA algorithm.

4. Train the generated multi-label dataset for the two tasks Sentiment Analysis and Topic Classification with five ML models, namely SVM, NB, KNN and LR, the DL model (LSTM), and LLMs based on ChatGPT.

5. Test the dataset in predicting the two labels "Sentiment" and "Topic" of each tweet in the dataset's test set.

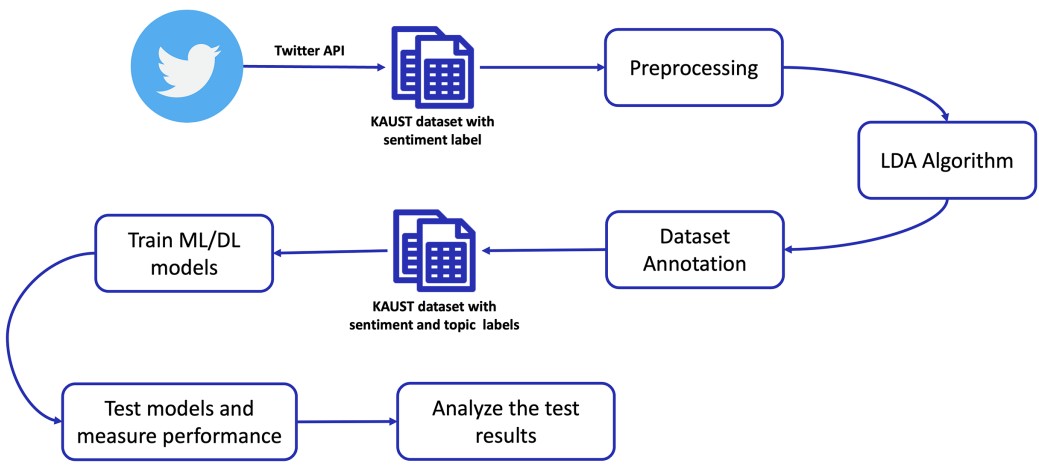

**Figure 3  Proposed research methodology framework.**

6. Analyze the test results based on the experiments' F1-score, CV result, and Bagging results to improve the dataset's performance.

This work analyzes the generated multi-label Arabic dataset to classify MSA tweets using topic-based sentiments. The experiments consist of two tasks: topic classification and sentiment analysis. To evaluate, we employed ML, DL, and LLMs to classify the topic and sentiment of an Arabic text with a series of data-processing techniques. The proposed approach of this article and the experiments were conducted by generating a multi-label dataset to build a topic-based sentiment analysis framework using machine learning models. Manually identifying each post's related topic and sentiment is unreliable and time-consuming. Hence, the need for a classifier capable of predicting the sentiment with the related topic of each Arabic tweet with high performance would assist the applications of Arabic NLP in various fields. The experimental outcomes highlight the importance of appropriate data preparation, including data cleaning, stemming, and vectorization. They also highlight the importance of employing an efficient weighting approach, such as TF-IDF and balancing data distribution across classes. These methods are implemented and tested with ML and DL models to determine which can produce the best results. The experiment has been done using Python 3.7 on the Google Colab framework, with the application of Keras API for DL (*Keras Team, 2021*), which is a Python-based high-level neural network API that can be used with TensorFlow (*Blokdyk, 2018*). The sample-weighted F1-score (SW F1-score) and F1-score were used to assess the models' performance. The harmonic mean of precision (Pr.) and recall (Rec.) could be defined as the F1-score:

$$F1 - Score = \frac{2 * Precision * Recall}{Precision + Recall} \tag{1}$$

$$Precision = \frac{TP}{TP + FP} \tag{2}$$

$$Recall = \frac{TP}{TP + FN}, \tag{3}$$

where True Positive (TP) represents the number of true positives, False Positive (FP) stands for the number of false positives, and False Negative (FN) shows the number of false negatives.

## MACHINE LEARNING

To discover the most effective method for predicting the sentiment and topic of a sentence, several ML models and one DL model have been tested. Four ML models, SVM, NB, KNN, and LR, were tested on the Multi-label dataset. As shown in Table 3, we have demonstrated that SVM achieved the best performance for topic classification at 0.96. data. For the other classifiers, NB and LR achieved 0.95, and NB achieved 0.91, whereas KNN had the worst performance with a 0.81 F1-score.

In the sentiment analysis task, the same experiments were repeated three times due to the data imbalance problem: balancing the data distribution using the SMOTE algorithm as an over-sampling technique, under-sampling technique, and without balancing the dataset. In addition, other ML classifiers have been tested: NB, SVM, KNN, and LR. As shown in Table 4, we have demonstrated that SVM achieved the highest performance of 0.95 with the oversampled dataset. The highest obtained results for NB and LR were similar (0.93), whereas KNN had the lowest performance with a 0.80 F1-score with the under-sampled dataset.

We have illustrated that data balancing affects the performance of all classifiers at different rates. Therefore, LR achieves the best performance with an imbalanced dataset at approximately 0.72, and SVM performs the best with a balanced dataset at 0.953 with the oversampled balanced dataset.

Since it is possible to have an improper data split that leads to biased learning outcomes, cross-validation (CV) is believed to be the preferred way for dividing the dataset into training and testing sets. To address this, cross-validation has been used to estimate the data split for all trained models critically.

The dataset has been evaluated on the same ML models with different-sized sub-sets to demonstrate the performance change over dataset sizes. Six subsets with a size of 500–20,000 are created. SVM outperformed the other five subgroups with just a minor difference in performance across different subsets. Tables 5 and 6 show the results obtained by applying the CV and Bagging technique for topic classification and sentiment analysis tasks. Figures 4 and 5 show the performance measured by F1-score, CV, and the Bagging technique of all tested models for sentiment analysis with both imbalanced and balanced dataset with under-sampling approach.

## DEEP LEARNING

Within the scope of our study, we conducted experiments to explore the performance of deep learning models, specifically focusing on LSTM networks when applied to our dataset. These models have demonstrated remarkable capabilities in various NLP tasks.

The LSTM model has three hidden layers, while the number of epochs has been set to 30. The results show an overfitting observed by higher precision than recall value. To

**Table 3 Topic classification Using ML models'.**

| Model | Politics | | Business | | Health | | F1-score | SW F1-score |
|-------|------|------|------|------|------|------|----------|-------------|
| | Pr. | Rec. | Pr. | Rec. | Pr. | Rec. | | |
| NB | 0.81 | 0.83 | 0.85 | 0.84 | 0.94 | 0.94 | 0.91 | 0.906 |
| KNN | 0.91 | 0.46 | 0.87 | 0.39 | 0.79 | 0.99 | 0.81 | 0.781 |
| SVM | 0.97 | 0.87 | 0.96 | 0.90 | 0.95 | 0.99 | 0.96 | 0.955 |
| LR | 0.97 | 0.86 | 0.95 | 0.90 | 0.95 | 0.99 | 0.95 | 0.953 |

**Table 4 Sentiment analysis Using ML models' with Various data balancing techniques.**

| Balancing technique | Mod. | Positive | | Negative | | Neutral | | F1-score | SW F1-score |
|---------------------|------|------|------|------|------|------|------|----------|-------------|
| | | Pr. | Rec. | Pr. | Rec. | Pr. | Rec. | | |
| Imbalanced dataset | NB | 0.42 | 0.39 | 0.42 | 0.35 | 0.78 | 0.82 | 0.68 | 0.676 |
| | KNN | 0.61 | 0.17 | 0.51 | 0.22 | 0.74 | 0.95 | 0.71 | 0.666 |
| | SVM | 0.79 | 0.11 | 0.79 | 0.11 | 0.71 | 0.99 | 0.71 | 0.634 |
| | LR | 0.75 | 0.16 | 0.76 | 0.13 | 0.73 | 0.98 | 0.72 | 0.656 |
| Over-sampling | NB | 0.91 | 0.99 | 0.91 | 0.99 | 0.98 | 0.79 | 0.92 | 0.897 |
| | KNN | 0.74 | 0.98 | 0.64 | 0.99 | 0.97 | 0.14 | 0.70 | 0.417 |
| | SVM | 0.78 | 0.17 | 0.80 | 0.11 | 0.73 | 0.99 | 0.97 | 0.657 |
| | LR | 0.94 | 0.95 | 0.94 | 0.95 | 0.91 | 0.89 | 0.93 | 0.914 |
| Under-sampling | NB | 0.60 | 0.63 | 0.61 | 0.62 | 0.62 | 0.59 | 0.60 | 0.608 |
| | KNN | 0.91 | 0.46 | 0.87 | 0.39 | 0.79 | 0.99 | 0.80 | 0.786 |
| | SVM | 0.61 | 0.72 | 0.63 | 0.66 | 0.72 | 0.57 | 0.64 | 0.641 |
| | LR | 0.61 | 0.71 | 0.61 | 0.64 | 0.70 | 0.56 | 0.63 | 0.628 |

**Table 5 Topic classification using CV and bagging.**

| Model | CV-score | BAGGING-score |
|-------|----------|---------------|
| NB | 0.909 | 0.907 |
| KNN | 0.804 | 0.799 |
| SVM | 0.953 | 0.952 |
| LR | 0.951 | 0.951 |

resolve overfitting, the number of epochs has been increased to 20 with an early stopping condition for better run-time optimization and to avoid overfitting caused by overtraining.

For the task of topic classification, the LSTM model exhibited a commendable F1-score of 0.93. This result highlights the model's capacity to identify complex patterns within the dataset and the efficacy of approaches in identifying and classifying text data into various topics.

LSTM achieved the highest F1-score of 0.96 for the sentiment analysis task by applying an over-sampling technique. This outcome signifies the importance of addressing the class

**Table 6 Sentiment analysis using CV and bagging.**

| Balancing technique | Model | CV-score | BAGGING-score |
|---|---|---|---|
| Imbalanced dataset | NB | 0.683 | 0.703 |
| | KNN | 0.683 | 0.703 |
| | SVM | 0.725 | 0.728 |
| | LR | 0.733 | 0.737 |
| Over-sampling | NB | 0.913 | 0.914 |
| | KNN | 0.687 | 0.710 |
| | SVM | 0.687 | 0.710 |
| | LR | 0.882 | 0.886 |
| Under-sampling | NB | 0.579 | 0.612 |
| | KNN | 0.801 | 0.805 |
| | SVM | 0.626 | 0.635 |
| | LR | 0.629 | 0.648 |

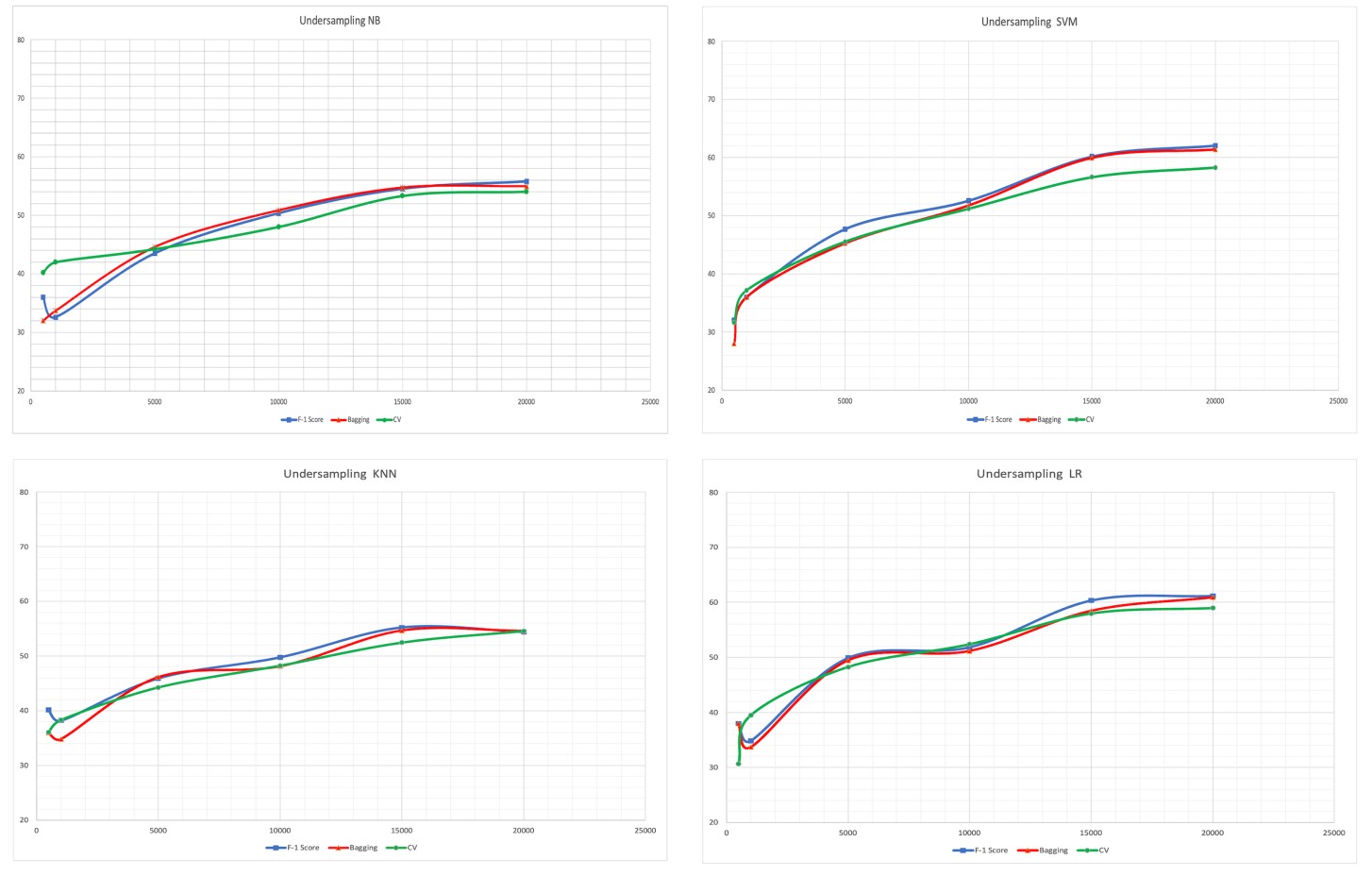

**Figure 4 Sentiment analysis performance with under-sample dataset.**

**Peer**J Computer Science

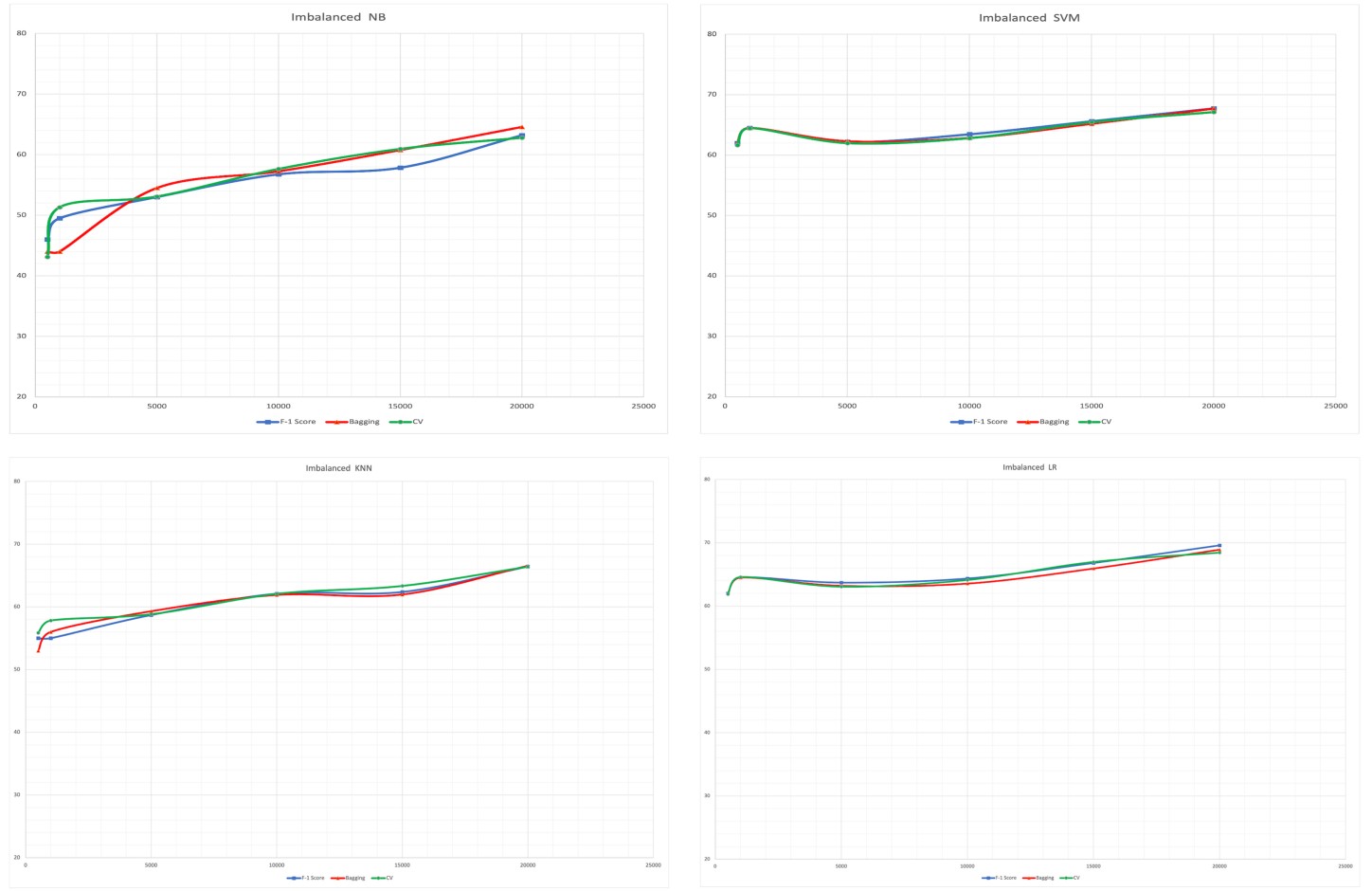

**Figure 5 Sentiment analysis performance with imbalanced dataset.**

**Table 7 DL based on LSTM model results.**

| Task | Support | F1-score | Standard deviation (s) |
|---|---|---|---|
| Topic classification | 6,552 | 0.93 | 0.05410 |
| Sentiment analysis (imbalanced data) | 13,678 | 0.93 | 0.08445 |
| Sentiment analysis (over sampling) | 13,678 | 0.96 | 0.09691 |
| Sentiment analysis (under sampling) | 13,678 | 0.69 | 0.07044 |

imbalance in sentiment analysis tasks, as it significantly improved the model's ability to recognize and classify sentiments accurately.

To provide a comprehensive overview of our findings, Table 7 summarizes the results obtained through DL for sentiment analysis and topic classification. Additionally, we have included the standard deviation of the experimental results, emphasizing the stability and consistency of our findings across multiple trials.

## GENERATIVE AI

In recent years, there has been a growing interest in using machine learning algorithms, including Large Language Models (LLMs), to automate the process of Topic-sentiment classification. Recently, ChatGPT, a pre-trained language model based on the Generative Pre-trained Transformer (GPT) architecture, is considered one of the most advanced LLMs for NLP applications. ChatGPT is a revolutionary chatbot platform developed by OpenAI that utilizes the latest advancements in AI and ML to provide a seamless conversational experience for users. One of the key features of ChatGPT is its advanced NLP capabilities. This allows it to understand and interpret user input more accurately and nuancedly, leading to more effective communication and problem-solving. It utilizes a transformer-based model to analyze text data. The model is trained on a large *corpus* of text, allowing it to recognize common patterns and relationships across many different types of text. This makes it highly effective at tasks like sentiment classification, where understanding the context and meaning of words is crucial, in addition to its ability to handle the complexity and nuance of the Arabic language (*OpenAI, 2023*). Since ChatGPT is a pre-trained LLM, it requires less training data than other ML algorithms, making it easier and faster to implement on various applications (*Feedback, 2022*). To that end, we have experimented ChatGPT API on our Arabic dataset to analyze the performance compared to the ML approaches.

### GPT-3.5

We have experienced GPT-3.5 API with *Google Sheets (2023)* since it supports the usage of ChatGPT through App Scripts add-ons. We run two commands asking, "The sentiment of "the tweet" is:" and "The general topic of "the tweet" is (classify it to one of the three topics: politics, business or health):" over our 500 tweets from our Arabic dataset to classify both classes sentiment and topic. The experiment was performed on a subset of the dataset because GPT-3.5 limited from processing the dataset at once, due to request time cap and processing timeout. GPT-3.5 was able to predict the text sentiment with 44.16% accuracy and 41.98% accuracy for the topic class on our Arabic Twitter dataset. We have observed that it could not assign a topic to the text in many cases due to the language. However, it was mostly able to assign a sentiment even with false predictions. As the platform currently supports GPT-3.5, it shows limited support for the Modern Standard Arabic language and cannot assign one of the specified topics to a text.

### GPT-4

GPT-4 is the 4th generation of the OpenAI language model. It offers enhanced accuracy, steerability, and faster processing due to the increased number of model parameters, (100 Trillion), which makes it 500 times faster than GPT-3.5, which has 175 billion parameters. It is currently available with an OpenAI Plus subscription. To measure the model accuracy on our Arabic Twitter dataset, we run the same two commands as done with GPT-3.5 on the same 500 tweets subset. GPT-4 was able to predict the text sentiment with 60.20% accuracy and 44.75% accuracy for the topic class. Even though the classification accuracy is

still low, it has been noticed that GPT-4 has a better ability to assign one of the predefined labels to a text than is with GPT-3.5.

## RESULTS DISCUSSION

The DL models are famous for their performance on complex data types, such as images, audio files, or large datasets with large feature space. In our case, the dataset is restricted to tweets related to COVID-19 on three topics. Such a specification, along with the short sentences of Tweets, leads to minimizing the feature space, reducing the required model complexity. This was proved in the literature that the ML approaches (NB and SVM) are more suitable for the case of analyzing Twitter data (*Boudad et al., 2018*; *Mostafa, 2017*; *Kharde & Sonawane, 2016*). Moreover, we found that balancing the dataset with oversampling techniques improves performance. The same argument was also supported in *Shwartz (2021)*, where the research survey showed that ML methods have the highest accuracy and can be considered baseline learning methods for Sentiment Analysis. These results prove that traditional ML classifiers can perform better than DL without overfitting the data when the dataset is not too large, as we have in our case, and the same has been proven in *Blokdyk (2018)* when they performed sentiment analysis on an English language dataset of size 30K tweets, a traditional ML algorithm (SVM) had the highest performance compared to other approaches, namely: sentiment lexicons, Off-the-shelf sentiment analysis systems such as Stanford CoreNLP system and DL algorithms. These results illustrate that SVM achieved the highest performance when applied for sentiment analysis and topic classification. This also proved that traditional ML approaches such as NB and SVM are more suitable for Twitter than the DL approaches. The DL once are much more convenient for complex datasets, such as images than Twitter, which contains short sentences and consists of smaller feature space.

The above experiments show that ML and ChatGPT approaches have advantages and disadvantages for topic-based sentiment classification. ML requires labeled data and can handle complex relationships between topics and sentiment but may not perform well on out-of-domain or previously unseen data. ChatGPT does not require labeled data and can generate responses for any topic but may not perform as well on complex or nuanced topics.

There could be several reasons why GPT gave lower accuracy with our Arabic language tweets dataset:

1. Lack of training data: GPT was trained on a large *corpus* of English language text. It may not have been trained on sufficient Arabic language data. This could cause lower accuracy when classifying Arabic language tweets.

2. Differences in language structure: Arabic is a Semitic language written from right to left, whereas English is a Germanic language written from left to right. The grammatical structure and vocabulary of Arabic are also different from English. These differences in language structure could make it more challenging for GPT to classify Arabic language tweets accurately.

3. Limited understanding of cultural references: GPT might need a greater understanding of cultural references and context-specific to Arabic language tweets. This could result in misinterpretation of the tweet content and inaccurate classification.

4. Lack of fine-tuning: GPT is a pre-trained language model that can be fine-tuned for specific tasks. Fine-tuning the model on Arabic language data could improve its accuracy (*Kheiri & Karimi, 2023*).

5. Data quality issues: One should note that data balancing has been performed while testing ML and DL models, whereas no data preprocessing has been applied with ChatGPT experiments. The quality of the Arabic language tweet data used for training and testing the model could impact its accuracy. If the data contains errors, noise, or bias, it could negatively affect the model's performance.

Ultimately, these approaches depend on the application's specific needs. Machine learning may be a better choice if a large amount of labeled data is available. If the topics are more diverse or there is no labeled data available, ChatGPT may be a better choice. Improving the performance could be done with different methods, like increasing the quality and quantity of the data and fine-tuning the model for the specific task using relevant data.

Figure 6 and Table 8 show the results of English and Arabic datasets compared to our 32 K Arabic Tweets dataset to compare the obtained performance on both languages with different dataset sizes, as well as the results of ChatGPT on our Arabic Tweets dataset. We can see that traditional ML classifiers achieved better results on smaller datasets while DL performs better on larger datasets for both languages.

## FUTURE DIRECTION

Building on our promising results with ChatGPT in understanding and analyzing the Arabic language, our future work will further push the boundaries of Arabic NLP and topic-based sentiment analysis. The current study lays a strong foundation, demonstrating ML, DL, and LLMs' proficiency in handling the nuanced complexities of Arabic and its applicability across various topics. However, recognizing NLP's dynamic and ever-evolving landscape, our future research direction includes several strategic expansions and enhancements.

In addition to the LDA approach, we have experimented with two other topic modeling algorithms, Bidirectional Encoder Representations from Transformers (BERT) and Non-Negative Matrix Factorization (NMF). Figure 7 shows the inter-topic distance map produced by BERT. It has shown promising results in visualizing the separability and coherence of topics. Additionally, the coherence value achieved by NMF was 0.1146, representing a different approach to topic modeling that could complement our findings with LDA. We aim to enhance the reliability of topic-based sentiment classification by advancing LLMs and improving current models for MSA. By exploring various algorithms and methodologies, such as Bert and NMF, and further refining LDA, we seek to develop more sophisticated and accurate tools for topic detection and sentiment analysis in Arabic.

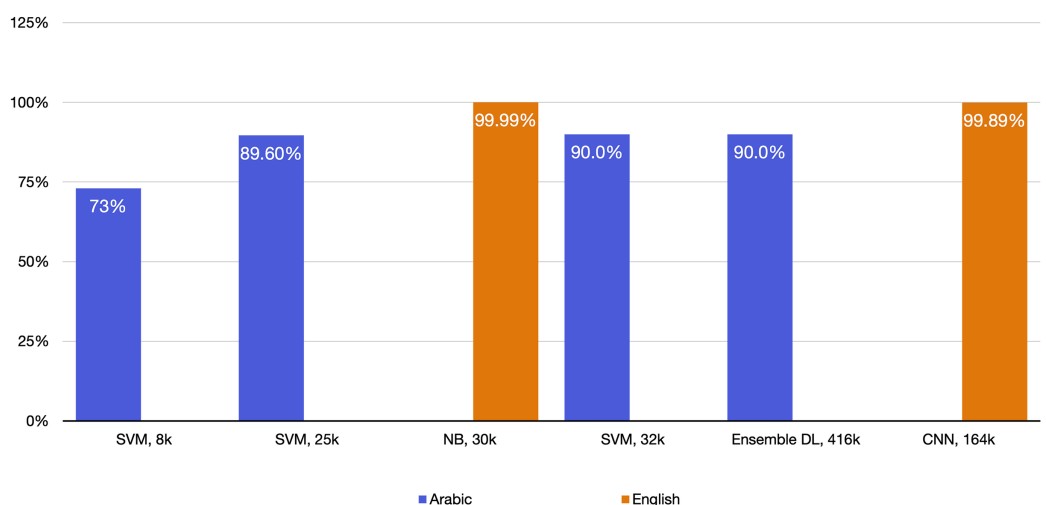

**Figure 6** Arabic and English datasets performance comparison with ML and DL approaches.

**Table 8 Performance results for Arabic and English datasets with ML, DL, and GPT approaches.**

| Dataset | Size | Language | Approach | Task | Highest performance |
|---|---|---|---|---|---|
| Twitter US airline sentiment (twi) | 30,000 | English | ML, DL, sentiment lexicons, and Off-the-shelf sentiment analysis systems | Sentiment analysis | NB achieved 73% accuracy |
| Prediction of user sentiment on the COVID-19 pandemic using tweets (*Mostafa, 2017*) | 1,646,004 | English | Several ML and DL Models | Sentiment analysis | CNN model achieved 99.89% accuracy |
| Tweets related to COVID-19 (*Alassaf & Qamar, 2022*) | 8,000 | Arabic | Several ML classifiers | Sentiment analysis | SVM achieved 85.4% accuracy |
| Tweets related to COVID-19 in Saudi Arabia (*Kharde & Sonawane, 2016*) | 416,292 | Arabic | SVM and an ensemble technique of three DL models: AraBert, SBGRU, and SGRU | Sentiment analysis | The ensemble model achieved 90% accuracy. |
| Multi-label dataset in online social networks (*Shwartz, 2021*) | 25,000 | Arabic | Several ML classifiers | Sentiment analysis | SVM achieved 89.6% accuracy |
| Multi-label Arabic COVID-19 tweets (*Alderazi, Algosaibi & Alabdullatif, 2022*) | 32,769 | Arabic | ML classifiers and LSTM | Topic-based sentiment classification | SVM achieved sentiment: 0.97 F1-score, topic: 0.96 F1-score |
| Multi-label Arabic COVID-19 tweets (*Alderazi, Algosaibi & Alabdullatif, 2022*) | 500 | Arabic | ML classifiers and LSTM | Topic-based sentiment classification | SVM achieved sentiment: 0.92 F1-score, topic: 0.92 F1-score |
| Multi-label Arabic COVID-19 tweets (*Alderazi, Algosaibi & Alabdullatif, 2022*) | 500 | Arabic | GPT-3.5 | Topic-based sentiment classification | Sentiment: 44.16% accuracy, topic: 41.98% accuracy |
| Multi-label Arabic COVID-19 tweets (*Alderazi, Algosaibi & Alabdullatif, 2022*) | 500 | Arabic | GPT-4 | Topic-based sentiment classification | Sentiment: 60.20% accuracy, topic: 44.75% accuracy |

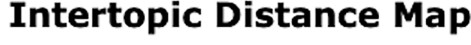

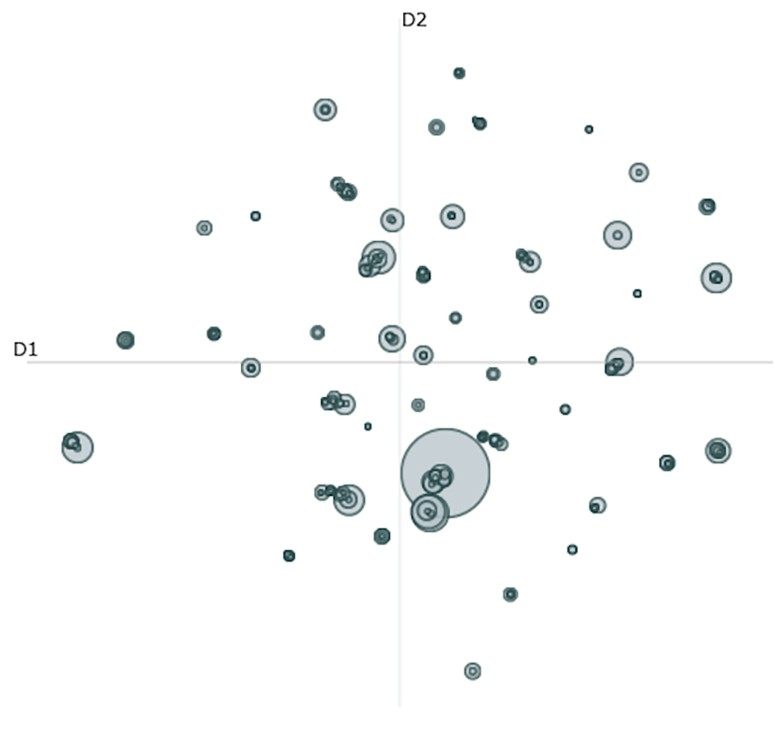

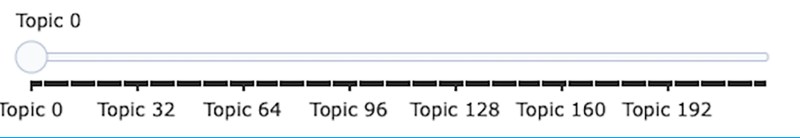

Figure 7 Bert inter-topic distance map.               

These efforts are expected to contribute significantly to the field of NLP, providing deeper insights and more reliable analyses of Arabic textual data.

However, recognizing NLP's dynamic and ever-evolving landscape, our future research direction includes several strategic expansions and enhancements.

1. Expansion of dataset and topics:
   We plan to broaden our dataset significantly, incorporating a larger and more contemporary dataset of Arabic texts. This expansion will provide a more robust base for analysis and help capture the evolving nuances and dialectical variations within the Arabic language. Additionally, increasing the variety and number of topics covered in our analysis will add depth and diversity to our understanding of sentiment across different domains. This expansion is crucial for developing more versatile and practical NLP tools.

2. Advancing Generative AI for dataset annotation:
   The potential of Generative AI to serve as an annotation tool is immense. In our subsequent phases, we will explore its utilization, especially GPT. This approach has

unique strengths in pattern recognition, handling non-linearity, and model generalization, which could be pivotal in dissecting the complexities of sentiment and topic labeling in Arabic texts.

3. Implementation of Ensemble Algorithms:
   Ensemble methods, which combine predictions from multiple ML algorithms to produce more accurate results than any individual model, will be another area of focus. By leveraging various algorithms' strengths and mitigating their weaknesses, ensemble methods can significantly boost the performance of topic-based sentiment analysis models.

4. Cross-linguistic comparisons:
   An intriguing extension of our research will be comparing Arabic sentiment analysis with other languages, particularly non-Indo-European languages with rich linguistic structures. This comparative analysis will not only aid in understanding the language-specific challenges in NLP but also pave the way for developing universal and adaptable topic-based sentiment analysis tools.

5. Addressing complex text analysis:
   While our current model exhibits competence in handling a wide range of texts, complex and nuanced expressions in Arabic still present a challenge. Future research will delve deeper into resolving ambiguities and improving understanding of context, idioms, and culturally specific expressions.

## CONCLUSIONS

To support enterprises' growth and research developments through social media data learning and to advance the field of Arabic NLP, this research contribution is centered on generating a Topic-Sentiment Classification approach for Arabic tweets. To build this topic-based sentiment analysis framework, several machine learning and deep learning models were analyzed on our generated multi-labeled Arabic Twitter dataset. This seeks to combine topic classification and sentiment analysis techniques. We have tested two data balancing techniques to overcome the imbalanced dataset challenge and applied cross-validation and ensemble techniques. The LSTM model and four additional machine learning classifiers—SVM, NB, KNN, and LR—were used to examine the dataset. With SVM surpassing all other classifiers in finding sentiment with 0.97 f1-score and topic classes with 0.96 f1-score, we demonstrated that using ML approaches with a balanced dataset through oversampling produced better outcomes than DL outcomes, which achieved an f1-score of 0.96 and 0.93 on classifying the sentiment and topic classes, respectively. In addition, Arabic topic-based sentiment classification using ChatGPT is a powerful tool for analyzing the sentiment of Arabic text. We have demonstrated its ability to handle the complexity of the Arabic language, and its topic-based approach makes it highly effective for a wide range of applications. As the field of natural language processing continues to evolve, we can expect to see even more advanced and sophisticated approaches to Arabic NLP. Although there are some challenges to analyze complex Arabic text, ChatGPT remains one of the most effective and efficient emerging tools. We plan to

work on a more extensive and more recent dataset in the future. Moreover, we intend to increase the number of topics to bring diversity to our research. Furthermore, a comparison with languages other than English could help identify language-specific challenges and opportunities.

### Funding
This work was funded by the Deanship of Scientific Research, King Faisal University, under project number: GRANT1,215. The funders had no role in study design, data collection and analysis, decision to publish, or preparation of the manuscript.

### Grant Disclosures
The following grant information was disclosed by the authors:
Scientific Research, King Faisal University: GRANT1,215.

### Competing Interests
The authors declare that they have no competing interests.

### Author Contributions
- Fatima Alderazi conceived and designed the experiments, performed the experiments, analyzed the data, performed the computation work, prepared figures and/or tables, authored or reviewed drafts of the article, and approved the final draft.
- Abdulelah Algosaibi conceived and designed the experiments, performed the experiments, analyzed the data, performed the computation work, prepared figures and/or tables, authored or reviewed drafts of the article, and approved the final draft.
- Mohammed Alabdullatif conceived and designed the experiments, performed the experiments, analyzed the data, prepared figures and/or tables, authored or reviewed drafts of the article, and approved the final draft.
- Hafiz Farooq Ahmad conceived and designed the experiments, analyzed the data, authored or reviewed drafts of the article, and approved the final draft.
- Ali Mustafa Qamar conceived and designed the experiments, performed the experiments, analyzed the data, prepared figures and/or tables, authored or reviewed drafts of the article, and approved the final draft.
- Abdulaziz Albarrak conceived and designed the experiments, authored or reviewed drafts of the article, and approved the final draft.

### Data Availability
   The code and data are available at GitHub and Zenodo:
   - https://github.com/ifatima95/Arabic_NLP.
   - Fatima. (2024). ifatima95/Arabic_NLP: final version (V0.1). Zenodo. https://doi.org/10.5281/zenodo.11114043.
   The code is available in the Supplemental Files.

## Supplemental Information

Supplemental information for this article can be found online at http://dx.doi.org/10.7717/peerj-cs.2081#supplemental-information.

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
