# Peer review of "Generative artificial intelligence in topic-sentiment classification for Arabic text: a comparative study with possible future directions"

_PeerJ Computer Science, doi:10.7717/peerj-cs.2081_

## Round 0.1 · original submission · Major Revisions

We have completed the 1st round review of your manuscript. You are required to address all the concerns and suggestions by reviewers and resubmit it. The revised manuscript will be subjected to a 2nd round review.

All 3 reviewers have major concerns and in particular, the serious concerns from R3 should be carefully and completely addressed.

Reviewer 1 ·

Basic reporting

This is a survey of natural language processing and machine learning about Arabic language. The authors utilize machine learning and NLP methods to classify the sentiment shared publicly into different topic models. The main contribution of this article is generating multilabel dataset with several classifiers.

Experimental design

The authors say the SVM achieve the highest performance as 0.97, and 0.96 f1-score in classifying the sentiment and topic of Arabic Tweets among all experiments. But it is the result of the application of different tools on the same database. What if we compare it to the previous algorithms?
The size of Multilabel Arabic COVID-19 tweets (13) in Table 7 is small, is it appropriate to be used as a standard dataset?

Validity of the findings

How about the relationship between Table 5 and Figure 4?

Reviewer 2 ·

Basic reporting

no comment

Experimental design

In this proposed work entitled “Generative artificial intelligence in topic-sentiment classification for Arabic text: A comparative study with possible future directions”, the authors evaluate multiple machine learning models and compare them with ChatGPT results for sentiment classifications on various topics in Arabic text. This work has shown some significant contributions and could be helpful for researchers working in the field of natural language processing and sentiment analysis. However, we have some comments to improve the present form of the manuscript:
1. The dataset among the class is imbalanced in both groups (Topic and Sentiment) and the authors suggested the oversampling/undersampling techniques used for balancing the dataset in this work. However, SMOTE (Synthetic Minority Oversampling Technique) [1] should be used or compared to eliminate any bias in the results.
2. Authors should analyse the result of sample-weighted F1 score is ideal for computing the net-F1 score for class-imbalanced data distribution.

References
1. https://dl.acm.org/doi/10.5555/1622407.1622416.

Validity of the findings

no comment

Additional comments

Authors need to proofread Table 7 and ensure the citation is made carefully.

Reviewer 3 ·

Basic reporting

1. The paper is well designed.
2. There are grammatical errors, etc. These should be fixed. There is no problem in terms of understanding the language of paper.
3. There is a misuse of LDA in line 80. Latin Dialect Allocation (LDA) should be corrected as Latent Dirichlet Allocation.
4. Because of LDA usage in topic modelling (TM) part of the study, literature studies are not novel. With the usage of novel techniques related to TM, the literature studies should be modified.
5. The scope of the study is not clear. It is not clear whether the study focuses on topic modeling or investigates the effect of class balancing techniques.
6. Here, class labels are compared by performing topic modeling on a dataset with sentiment information. If so, what is the purpose of using machine learning? Or is this a measure of how consistent the class information obtained through topic modeling is with the labels in the already labeled dataset? At the end of the day, is the emphasis on the performance of labeling unlabeled data using topic modeling? These parts are not clearly detailed.
7. Is there any novelty, this should be clearly defined. It seems just the experimentation of different ML model on the dataset.
7. Figure 3 is blurred. It is hard interpret because of the quality of the Figure 3.

Experimental design

1. Research question is defined but not sufficient. The exact focus of the article was not given. Therefore, It's hard to connect it to results.

2. LDA is a very old topic modeling technique. Why is the work based only on LDA when there are innovative transformer-based approaches (Bertopic, Top2vec, etc..) that perform better?

3. I couldn't see any details about how the LDA model was implemented. Did you work with uni-grams? So why?

4. Evaluation metrics showing that LDA is successful in topic modeling are not included. Coherent score, perplexity etc.

5. I could not find any information about the parameter details of the machine learning algorithms used. Was parameter optimization performed in this study, and if so, what results were obtained with what values?

6. It would be good to add English equivalents to Arabic texts to show how consistent any results are in the Wordcloud visual, results, and text content.

7. Using f1-score as an evaluation metric to evaluate results is insufficient. Especially, in cases where there is class imbalance in the data set, it is necessary to expand these metrics such as precision, recall, sensitivity and specificity. Because these metrics provide information about class imbalance in the data set. The results of these metrics should be compared before and after balancing techniques are applied.

8. Parameter details are not given when using deep learning techniques. Additionally, there is no evidence whether the model exhibits over-fitting problem or not. There is no information about whether fine tuning or parameter optimization was performed while applying the model. In this respect, it is difficult and unrealistic to evaluate the success of the model.

Validity of the findings

It is not clear what purpose the article serves. The content seems to implement a traditional sentiment classification task using traditional methods. I don't see any innovation in this respect. It is difficult to say that it contributes to the literature when basic problems such as the lack of details in the models and whether the models have over-fitting problems are evaluated.

On the other hand, No information has been provided to prove the accuracy of the results.

---

## Round 0.2 · Minor Revisions

Dear authors,
Reviewer #3 still has some concerns about the revised manuscript. Please check each comments of the Reviewer 3, prepare changes, and upload a point-by-point rebuttal.

Reviewer 2 ·

Basic reporting

No comments

Experimental design

No comments

Validity of the findings

No Comments

Additional comments

No Comments

Reviewer 3 ·

Basic reporting

The authors did not clearly write or emphasize the purpose, motivation, and innovation of the article requested for revision in the abstract section.

Experimental design

.

Validity of the findings

.

Additional comments

Authors should address in the reply file to the referees which changes were made, line by line, etc. I observed that some of the requested changes were made in the entire article and some were not. In order not to be overlooked, authors should notify the changes they have made by giving the chapter name and line order.

---

## Round 0.3 · accepted · Accept

I am pleased to inform you that your paper has been accepted for publication in PeerJ Computer Science. Your manuscript has undergone two-rounds of rigorous peer review, and I am delighted to say that it has been met with high praise from our reviewers and editorial team. On behalf of the editorial board, I extend our warmest congratulations to all the contributors.